# Quantifying Character Similarity with Vision Transformers

**Xinmei Yang[1], Abhishek Arora[1], Shao-Yu Jheng[1], Melissa Dell[1,2*]**

[1]Harvard University; Cambridge, MA, USA.

[2]National Bureau of Economic Research; Cambridge, MA, USA.

[*]Corresponding author: melissadell@fas.harvard.edu.

## Abstract

Record linkage is a bedrock of quantitative social science, as analyses often require linking data from multiple, noisy sources. Off-the-shelf string matching methods are widely used, as they are straightforward and cheap to implement and scale. Not all character substitutions are equally probable, and for some settings there are widely used handcrafted lists denoting which string substitutions are more likely, that improve the accuracy of string matching. However, such lists do not exist for many settings, skewing research with linked datasets towards a few high-resource contexts that are not representative of the diversity of human societies. This study develops an extensible way to measure character substitution costs for OCR'ed documents, by employing large-scale self-supervised training of vision transformers (ViT) with augmented digital fonts. For each language written with the CJK script, we contrastively learn a metric space where different augmentations of the same character are represented nearby. In this space, homoglyphic characters - those with similar appearance such as "O" and "0" - have similar vector representations. Using the cosine distance between characters' representations as the substitution cost in an edit distance matching algorithm significantly improves record linkage compared to other widely used string matching methods, as OCR errors tend to be homoglyphic in nature. Homoglyphs can plausibly capture character visual similarity across *any* script, including low-resource settings. We illustrate this by creating homoglyph sets for 3,000 year old ancient Chinese characters, which are highly pictorial. Fascinatingly, a ViT is able to capture relationships in how different abstract concepts were conceptualized by ancient societies, that have been noted in the archaeological literature.

## 1 Introduction

Many quantitative analyses in the social sciences - as well as government and business applications - require linking information from multiple datasets. For example, researchers and governments link historical censuses, match hand-written records from vaccination campaigns to administrative data, and de-duplicate voter rolls. The sources to be linked often contain noise, particularly when they were created with optical character recognition (OCR). String matching methods are widely used to link entities across datasets, as they are straightforward to implement off-the-shelf and can be scaled to massive datasets (Binette and Steorts, 2022; Abramitzky et al., 2021).

Most simply, approximate string matching methods count the number of edits (insertions, deletions, and substitutions) to transform one string into another (Levenshtein et al., 1966). Another common approach computes the similarity between $n$-gram representations of strings, where $n$-grams are all substrings of length $n$ (Okazaki and Tsujii, 2010).

In practice, not all string substitutions are equally probable, and efforts to construct lists that vary their costs date back over a century. For example, in 1918 Russell and Odell patented Soundex (Russell, 1918; Archives and Administration, 2023), a sound standardization toolkit that accounts for the fact that census enumerators often misspelled names according to their sound. Together with the updated New York State Identification and Intelligence System (Silbert, 1970), it remains a bedrock for linking U.S. historical censuses (Abramitzky et al., 2021). Similarly, Novosad (2018) adjusts Levenshtein distance to impose smaller penalties for common alternative spellings in Hindi, and the FuzzyChinese package (znwang25, 2020) uses strokes as the unit for $n$-grams substring representations, where the strokes for a given character are drawn from an external database (kfcd, 2015) covering a subset of the CJK script. Characters sharing strokes are more likely to be matched.

Such methods can perform well in the contexts for which they are tailored but are labor-intensive

to extend to new settings, due to the use of hand-crafted features. Low extensibility skews research with linked data - necessary to examine intergenerational mobility, the evolution of firm productivity, the persistence of poverty, and many other topics - towards a few higher resource settings that are not representative of the diversity of human societies.

This study aims to preserve the advantages of string matching methods - simple off-the-shelf implementation and high scalability - while developing an extensible, self-supervised method for determining the relative costs of character substitutions in databases created with OCR. OCR often confuses characters with their homoglyphs, which have a similar visual appearance (*e.g.* "0" and "O"). Incorporating character visual similarity into string matching can thus plausibly improve record linkage. Homoglyphs can be constructed by hand for small script sets such as Latin, as in a psychology literature on literacy acquisition (Simpson et al., 2013). For a script such as CJK, containing over 38,000 characters, this is infeasible.

Following a literature on self-supervision through simple data augmentation for image encoders (Grill et al., 2020; Chen et al., 2021; Chen and He, 2021), this study uses augmented digital fonts to contrastively learn a metric space where different augmentations of a character (*e.g.*, the character rendered with different fonts) have similar vector representations. The resulting space can be used, with a reference font, to measure the visual similarity of different characters. This purely self-supervised approach can be extended to any character set. Due to space constraints, this study focuses on languages written with CJK: Simplified and Traditional Chinese, Japanese, and Korean.

We train the HOMOGLYPH model on augmentations of the same character - rather than paired data across characters - because a self-supervised approach is more extensible. Paired character similarity data are limited. Unicode maintains a set of confusables - constructed with rule-based methods - but for CJK the only confusables are structurally identical characters with different Unicode codepoints. Despite a large post-OCR error correction literature (Lyu et al., 2021; Nguyen et al., 2021; van Strien. et al., 2020), there is also limited ground truth data about the types of errors that OCR makes across architectures, languages, scripts, layouts, and document contexts.

Using the cosine distance between two charac-ters as the substitution cost within a Levenshtein edit distance framework (Levenshtein et al., 1966) significantly improves record linkage, relative to other string matching methods. We first examine linking real world data on supply chains, which are central to a variety of economic questions. We use three very different open-source OCR architectures to digitize supply chain and firm information from two 1950s Japanese publications (Jinji Koshinjo, 1954; Teikoku Koshinjo, 1957). We then use HOMOGLYPH to link them, significantly improving linkage over other string matching methods. This exercise illustrates that OCR errors tend to be homoglyphic regardless of the OCR architecture used.

We provide evaluations for additional languages using synthetically generated data, as creating evaluation data is very costly. We augment image renders of place and firm names written with different fonts, for the Simplified and Traditional Chinese, Japanese, and Korean character sets. We then OCR two different views of each entity with different OCR engines. The different augmentations and OCR engines lead with high frequency to different text string views of the same entity. We then link these using string matching methods. Homoglyphic matching outperforms other widely used string matching techniques for all four languages.

While end-to-end deep neural methods could plausibly outperform string matching, the data required for them are not always available and technical requirements for implementation are higher, explaining why string matching methods predominate in social science applications. Homoglyphic matching is a cheap and extensible way to improve string matching. Our python package HomoglyphsCJK (https://pypi.org/project/HomoglyphsCJK/) provides a simple, off-the-shelf implementation.

Homoglyphs can be extended to any script. To explore this, we contrastively train a HOMOGLYPH model for ancient Chinese characters, using a database that provides views of the same character from different archaeological sites and time periods (Academia Sinica et al., 2023). Ancient characters are much more pictorial than their more abstract, modern equivalents. Fascinatingly, homoglyphs constructed with a ViT for the Shang Dynasty (1600 BC-1045 BC) capture ways in which ancient Chinese society related abstract concepts that have been noted in the archaeological literature

(*e.g.* Wang (2003)).

The rest of this study is organized as follows: Section 2 develops methods for learning character similarity and incorporating it into string matching, and Section 3 describes the evaluation datasets. Section 4 compares the performance of homoglyphic edit distance to other string matching methods for record linkage, and Section 5 introduces the HomoglyphsCJK package. Section 6 examines extensibility by constructing homoglyphs for ancient Chinese, and Section 7 discusses the limitations of homoglyphs.

## 2 Methods

### 2.1 The HOMOGLYPH model

The HOMOGLYPH model contrastively learns a mapping between character crops and dense vector representations, such that crops of augmentations of the same character are nearby, as illustrated in Figure 1. HOMOGLYPH is trained purely on digital fonts. Figure 2 shows variations of the same characters rendered with different fonts, which form positive examples for training.[1] Variations across fonts are non-trivial, forcing the model to learn character similarities at varying levels of abstraction.

We use a DINO (Self-**Di**stillation, **No** Labels) pre-trained ViT as the encoder (Caron et al., 2021). DINO ViT embeddings perform well as a nearest neighbor classifier, making them well-suited for homoglyphic matching. The model is trained using a Supervised Contrastive loss function (Khosla et al., 2020), a generalization of the InfoNCE loss (Oord et al., 2018) that allows for multiple positive and negative pairs for a given anchor:

$$\sum_{i \in I} \frac{-1}{|P(i)|} \sum_{p \in P(i)} \log \frac{\exp\left(\boldsymbol{z}_i \cdot \boldsymbol{z}_p / \tau\right)}{\sum_{a \in A(i)} \exp\left(\boldsymbol{z}_i \cdot \boldsymbol{z}_a / \tau\right)} \quad (1)$$

where $\tau$ is a temperature parameter (equal to 0.1), $i$ indexes a sample in a "multiviewed" batch (in this case multiple fonts/augmentations of characters with the same identity), $P(i)$ is the set of indices of all positives in the multiviewed batch that are distinct from $i$, $A(i)$ is the set of all indices excluding $i$, and $z$ is an embedding of a sample in the batch. Training details are describe in the supplementary materials.

To compute characters' similarity, we embed their image crops, created with a reference font (Google Noto) chosen for its comprehensiveness, and compute cosine similarity with a Facebook Artificial Intelligence Similarly Search backend (Johnson et al., 2019).

Figure 3 shows representative examples of characters and their five nearest homoglyphs. Characters with similar vector representations have qualitatively similar appearances.

HOMOGLYPH shares common elements with EfficientOCR (Carlson et al., 2023), an OCR architecture that learns to recognize characters by contrastively training on character crops rendered with augmented digital fonts. Different augmentations of a character provide positive examples. At inference time, localized characters are OCR'ed by retrieving their nearest neighbor from an offline index of exemplar character embeddings. EfficientOCR aims to retrieve the same character in an offline index, whereas HOMOGLYPH measures similarity across characters. While HOMOGLYPH shares the architecture of the EfficientOCR character recognizer, it does not use the same model weights or training data.

### 2.2 String Matching Methods

Dunn (1946) - in one of the first treatments of record linkage - wrote: "Each person in the world creates a Book of Life. This Book starts with birth and ends with death. Its pages are made up of the records of the principal events in life. Record linkage is the name given to the process of assembling the pages of this Book into a volume."

Edit distance metrics are widely used for this task *e.g.* Levenshtein et al. (1966); Jaro (1989); Winkler (1990). Another common approach computes the cosine similarity between $n$-gram representations of strings (Okazaki and Tsujii, 2010).

There are a variety of ways that character-level visual similarity could be incorporated into record linkage. We follow the literature modifying Levenshtein distance, e.g. Novosad (2018), by using cosine distance in the HOMOGLYPH space as the substitution cost. Insertion and deletion costs are set to one. It is straightforward to scale the insertion and deletion costs using parameters estimated on a validation set, but we focus on performance without any tuned parameters to maintain a purely off-the-shelf, self-supervised implementation.

We compare matching with homoglyphic edit

---

[1] We use 62 open-source fonts for Korean, 27 for Simplified Chinese, 17 for Traditional Chinese, and 13 for Japanese. Our training budget required us to base the number of fonts for each script on the total number of characters in that script, using more fonts for characters with a smaller Unicode range.

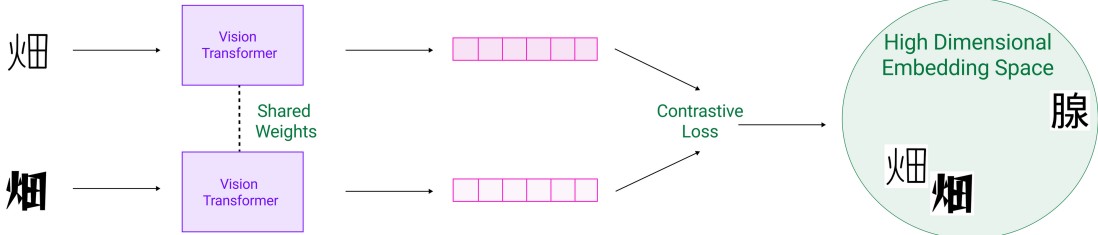

Figure 1: **Training Architecture**. Contrastive training architecture for the HOMOGLYPH model.

Figure 2: **Character variation across fonts.** This figure illustrates examples of the same character rendered with different fonts. Augmentations of these comprise positives in the HOMOGLYPH training data.

Figure 3: **Homoglyphs.** This figure illustrates the five nearest neighbors in the HOMOGLYPH embedding space for representative characters.

(kfcd, 2015) covering a subset of the CJK script.

## 3 Evaluation Datasets

To our knowledge, there are no widely used benchmarks for evaluating record linkage for the CJK script, and thus we develop datasets for evaluation. Table 1 contains information about their sizes.

First, we link a dataset on the customers and suppliers of major Japanese firms, drawn from a 1956 Japanese firm publication (Jinji Koshinjo, 1954), to a firm index of around 7,000 firms. The index is from the same publication but written in a different font. Firm names are localized with LayoutParser (Shen et al., 2021), using a custom layout analysis

distance to a variety of other methods. The first comparison is to classic Levenshtein distance (insertions, deletions, and substitutions are all equally costly), to isolate the effect of varying the substitution cost. We also compare to the popular Simstring package, which uses a variety of similarity metrics (Jaccard, cosine, and Dice similarity), computed with 2-gram substrings (Okazaki and Tsujii, 2010). The third comparison is to FuzzyChinese, a widely used package that uses strokes or characters as the fundamental unit for n-gram substring representations (we use the default 3-grams). These are compared using the TF-IDF vectors. The strokes in each character are drawn from an external database

| Task | Query | Target |
|---|---|---|
| Hist. Jap. Companies Across (*historic-ja-across*) | 238 | 68,352 |
| Hist. Jap. Companies Within (*historic-ja-within*) | 1,092 | 6,725 |
| Synthetic Japanese (*synth-ja*) | 86,470 | 86,470 |
| Synthetic Korean (*synth-ko*) | 48,809 | 48,809 |
| Synthetic Trad. Chinese (*synth-zht*) | 66,943 | 66,943 |
| Synthetic Simp. Chinese (*synth-zhs*) | 20,162 | 20,162 |

Table 1: Query (items to match) and target (reference dataset) sizes for each record linkage task.

model that detects individual customers and suppliers in lists. Custom layout analysis is necessary, as some OCR engines fail to detect the character separating firms in the customer-supplier list. We then hand-link a randomly selected sample of 1092 customers and suppliers to the index, keeping only one instance of each firm in the customer-supplier lists to create a diverse evaluation set. We call this dataset *historic-ja-within*.

Firm names are OCR'ed twice, to shed light on whether errors tend to be homoglyphic for both vision-only and vision-language sequence-to-sequence OCR architectures. We employ two widely used, open-source engines: EasyOCR and PaddleOCR. EasyOCR uses a convolutional recurrent neural network (CRNN) (Shi et al., 2016), with learned embeddings from a vision model serving as inputs to a learned language model. PaddleOCR abandons language modeling, dividing text images into small patches, using mixing blocks to perceive inter- and intra-character patterns, and recognizing text by linear prediction (Du et al., 2022). Neither engine localizes individual characters.

In a second exercise, we use a hand-curated dataset that links 238 firms to a much larger firm directory containing over 70,000 firms (Teikoku Koshinjo, 1957). This latter source is written vertically, which PaddleOCR and EasyOCR do not support. Instead, we digitize them with EfficientOCR (Bryan et al., 2023), custom-trained on both target publications. We would expect EfficientOCR's character retrieval framework to make homoglyphic errors. The large firm directory contains some firms with duplicate names; these are removed for the purposes of this analysis, as space constraints do not permit a discussion of varied blocking methods (Binette and Steorts, 2022).

Because creating ground truth data for record linkage is costly, we use synthetically generated data for a third set of evaluations. For Simpli-

fied Chinese (*synth-zhs*), Japanese (*synth-ja*), and Korean (*synth-ko*), we draw placenames from the Geonames database (Geonames, 2023). Because Traditional Chinese placenames in Geonames are rare, we instead draw from a list of Taiwanese firms (Taiwan Ministry of Economic Affairs, 2023), as Taiwan still uses Traditional Chinese (forming the dataset *synth-zht)*. We augment the images of each entity name, randomly select two image crops for each entity, and OCR them using EasyOCR and PaddleOCR. Anywhere from 40% (Simplified Chinese) to 88% (Traditional Chinese) of OCR'ed string pairs differ between the two OCR engines, and we use homoglyphic edit distance to match these strings.[2]

## 4    Results

Homoglyphic edit distance outperforms the other string matching methods in all three evaluation exercises - across different OCR engines and languages - typically by an appreciable margin. This illustrates that homoglyphic errors in OCR are common and can be captured with self-supervised vision transformers.

We show that homoglyphs are useful for real-world research tasks, starting with supply chains. Supply chain are widely studied, as they are fundamental to the transmission of economic shocks (Acemoglu et al., 2016, 2012), agglomeration (Ellison et al., 2010), and economic development (Hirschman, 1958; Myrdal and Sitohang, 1957; Rasmussen, 1956; Bartelme and Gorodnichenko, 2015; Lane, 2022). Yet their role in long-run economic development has been difficult to study due to the challenges of accurately linking large-scale historical records

Our first evaluation exercise, with linked Japanese supply chain data (*historic-ja-within*) - aims to elucidate whether homoglyphic matching is as helpful for linking datasets created with vision-language OCR (EasyOCR) as for linking datasets created with vision-only OCR (PaddleOCR), and whether it can similarly be useful for linking datasets created with different OCR architectures. We hence separately consider results linking PaddleOCR'ed customers and suppliers to the EasyOCR'ed firm index, and vice-versa, as well as linking when both are OCR'ed by either PaddleOCR

---

[2]The sample size is 20,162 for Simplified Chinese, 66,943 for Traditional Chinese, 86,470 for Japanese, and 48,809 for Korean.

| Method | OCR Engines | | | |
| | Paddle to Easy | Easy to Paddle | Paddle to Paddle | Easy to Easy |
|---|---|---|---|---|
| Homoglyphic distance | **0.808** | **0.753** | **0.844** | **0.728** |
| Levenshtein distance | 0.766 | 0.697 | 0.807 | 0.693 |
| Simstring (cosine) | 0.762 | 0.662 | 0.787 | 0.673 |
| Simstring (dice) | 0.763 | 0.663 | 0.788 | 0.673 |
| Simstring (jaccard) | 0.763 | 0.663 | 0.788 | 0.673 |
| FuzzyChinese (stroke) | 0.690 | 0.567 | 0.717 | 0.554 |
| FuzzyChinese (character) | 0.533 | 0.445 | 0.559 | 0.464 |

Table 2: **String Matching Across OCR Engines for** *historic-ja-within*. This table reports accuracy using a variety of different methods for linking Japanese firms from supply chain records (Jinji Koshinjo, 1954) to a firm index created from the same publication. The four columns report results when (1) PaddleOCR is used to OCR the firm list and EasyOCR the directory, (2) EasyOCR is used to OCR the firm list and PaddleOCR the directory, (3) PaddleOCR is used to OCR both lists, (4) EasyOCR is used to OCR both lists.

or EasyOCR. Homoglyphic edit distance outperforms other string matching methods and does so by a similar margin (around 4 percentage points higher accuracy) regardless of the OCR architecture used. FuzzyChinese has the weakest performance, as expected since many Japanese characters are not covered in their stroke dictionary.

Our second evaluation, *historic-ja-across* examines matching to a different, vertically written book, with around ten times more firms, digitized with EfficientOCR because it supports vertical Japanese. Homoglyphic distance outperforms all other string matching methods, with a matching accuracy of 82%. This matching exercise is also examined in Arora et al. (2023), using a customized, end-to-end multimodal deep neural model to link firms across the books. This achieves an accuracy of 87.8 with Vision-only contrastive training and a peak accuracy of 94.5 using a Language-Image model (CLIP) backbone with multimodal pooling. The end-to-end deep neural methods outperform string matching, as would be expected, as they are leveraging more information (the image crops as well as the texts) and language understanding. Yet string matching methods remain prevalent in the literature, due to their straightforward off-the-

| Method | Accuracy |
|---|---|
| Homoglyphic distance | 0.824 |
| Levenshtein distance | 0.731 |
| Simstring (cosine) | 0.748 |
| Simstring (dice) | 0.752 |
| Simstring (jaccard) | 0.752 |
| FuzzyChinese (stroke) | 0.735 |
| FuzzyChinese (character) | 0.618 |

Table 3: **Linking Japanese Firms to a Large Directory:** *historic-ja-across* This table links Japanese firms from supply chain records (Jinji Koshinjo, 1954) to an extensive firm directory (Teikoku Koshinjo, 1957), comparing various string matching methods.

| | synth-**ja** | synth-**ko** | synth-**zhs** | synth-**zht** |
|---|---|---|---|---|
| Homoglyphic distance | **0.456** | **0.292** | **0.476** | **0.465** |
| Levenshtein distance | 0.396 | 0.188 | 0.375 | 0.407 |
| Simstring (cosine) | 0.376 | 0.247 | 0.425 | 0.383 |
| Simstring (dice) | 0.380 | 0.248 | 0.426 | 0.385 |
| Simstring (jaccard) | 0.380 | 0.248 | 0.426 | 0.385 |
| FuzzyChinese (stroke) | 0.168 | 0.000 | 0.473 | 0.372 |
| FuzzyChinese (character) | 0.230 | 0.110 | 0.137 | 0.197 |

Table 4: **Matching Results: Synthetic Data**. This table reports accuracy linking synthetic paired data generated by OCR'ing location and firm names - rendered with augmented digital fonts - with two different OCR engines, four the four languages in out exercise.

shelf implementation, interpretability, familiarity to users, minimal input requirements (only OCR'ed strings), and the fact that language understanding - while relevant for linking firms - may be of less relevance for linking individuals, placenames, etc.

Finally, Table 4 reports results with the synthetically generated record linkage datasets, to elucidate the performance of homoglyphic matching across languages that use the CJK script. Homoglyphs outperform other string-matching methods. The only case where the performance of another method is similar is Simplified Chinese, with the FuzzyChinese package using stroke level $n$-grams. The stroke dictionary that underlies FuzzyChinese was crafted for Simplified Chinese, yet homoglyphs can perform similarly with self-supervised methods. On Traditional Chinese, which proliferates in historical documents, homoglyphic edit distance

| (1)
Ground
Truth
String | (2)
PaddleOCR
Recognition | (3)
EasyOCR
Recognition
(Correct
Match) | (4)
Homoglyphic
Match | (5)
Levenshtein
Match | (6)
Simstring
Cosine
Match | (7)
FuzzyChinese
Stroke
Match |
|---|---|---|---|---|---|---|
| Panel A: Homoglyphic Matching Error Cases | | | | | | |
| 有楽町 | 有楽田 | 有業町 | 有**馬日** | **馬**田 | 有**明**町**原**田 | **三**楽町 |
| 율치 | 율치 | 율기 | **운**치 | **백**치 | **가**치 | **용이** |
| 西阳呈 | 西阳呈 | 丏阳三 | **酉**阳**寺** | 西**沿湾** | 西阳**夕村** | **酉**阳**寺** |
| 陳 金 灯 | 陳金小 | 陳金燈 | 陳金**火** | 陳金**呈** | 陳金**)** | **王小**金 |
| Panel B: Homoglyphic Matching Correct Cases | | | | | | |
| たかだけ | た**カ**だ**ナ** | たかだけ | た**か**だ**け** | た**せ**だ**む** | た | **かでなひこう
じょう** |
| 양사동 | 양사동 | 양사**돈** | 양사**돈** | 양사**전** | **#**사동 | **용이** |
| 石涌 | 石涌 | 石**浦** | 石**浦** | 石**船** | **槎**涌 | **槎**涌 |
| 陳淑汝 | 陳淑**女** | 陳淑汝 | 陳淑**汝** | 陳淑**麗** | **曾**淑**女** | **林**淑**女** |

Figure 4: **Error analysis.** Panel A shows representative errors from homoglyphic matching. Panel B shows representative cases that homoglyphic matching gets correct. The ground truth string is shown in column (1). PaddleOCR is used to OCR the query images (column (2)) and EasyOCR is used to OCR their corresponding keys (column (3)). Columns (4) through (7) give the selected match to the query using different string matching methods, with the correct match shown in column (3). Bold characters differ from the query.

offers a nine percentage point accuracy advantage over FuzzyChinese, illustrating the extensibility advantages of self-supervised methods. The accuracy rates are low, but this must be interpreted in the context of the dataset, which only includes paired records where the OCR differs.

Figure 4 provides an error analysis for the synthetic record linkage exercise. The ground truth string, taken from the original image, is shown in the first column. PaddleOCR is used to OCR the query (column 2). EasyOCR is used to OCR the key, with the resulting string shown in column (3). The matches selected from the key by different string-matching methods are shown in columns (4) through (7). Bold characters differ from the query OCR. Panel A shows cases where homoglyphic edit distance selects an incorrect match. This typically occurs when the OCR'ed key has a similar visual appearance to an incorrect match in the query, showing the limits of homoglyphs to fully alleviate the OCR information bottleneck. Panel B shows cases where homoglyphic edit distance selects a correct match.

## 5 The HomoglyphsCJK package

We distribute the homoglyphic matching approach for CJK languages as a Python package [3] which has an API designed around the standard merge operation in pandas - effectively reducing the entry barrier to using it not only for Python users but also those experienced in using R or Stata for data analysis. All the user needs are the two datasets to match (with the key to match on) and one line of code. The package comes pre-loaded with the lookup tables that contain pairwise distances between different characters for a script. The files are also available on HuggingFace[4].

We provide two main functions in the package - one to calculate the homoglyphic distance (*hg_distance*) between a pair of strings and the other for matching two data frames (*hg_merge*). Here is example usage.

```
import pandas as pd
from HomoglyphsCJK import  hg_distance,
    hg_merge

df1=pd.read_csv("df1.csv")
df2=pd.read_csv("df2.csv")
df_merged = hg_merge('zhs',df1,df2,'
    query','key',homo_lambda=1,
    insertion=1, deletion=1)
```

```
hg_distance("太阳村","月亮湾",
'zhs',homo_lambda=1, insertion=1,
    deletion=1)
```

The distance calculations are fully customizable, with the weights of insertion and deletion tunable. "homo_lambda" allows the user to scale the homoglyphic substitution cost, thereby adding another dimension to tune the match quality. (In the results

---

[3]https://pypi.org/project/HomoglyphsCJK/

[4]https://huggingface.co/datasets/dell-research-harvard/HomoglyphsCJKTraining

above, we use the default of setting these all to 1.) The package repo contains more documentation and a colab notebook to allow quick exploration of the package's functionality.[5] For users who are interested in computing their own character visual similarity indices, we provide our training code in a separate repo.[6]

## 6 Extending Homoglyphs

While this study focuses on the modern CJK script, HOMOGLYPH can be extended to any character set. As a proof of concept, we explore its extensibility to ancient Chinese characters. Like other early forms of human writing, ancient Chinese scripts are highly pictorial relative to modern characters.

Since we do not have multiple fonts for ancient Chinese characters, we instead use an existing database of grouped ancient characters - from different archaeological sites and periods - that correspond to the same concept (Academia Sinica et al., 2023). These ancient characters are also linked to their descendant modern (Traditional) Chinese character, identified by the database. We contrastively learn a metric space where the representations of ancient characters denoting the same concept are nearby. We train on 25,984 ancient characters, as well as the corresponding augmented Traditional Chinese characters. The dataset includes characters from the Shang Dynasty (1600 BC-1045 BC), the Western Zhou (1045 BC-771 BC), the Spring and Autumn Warring States Era (770 BBC -221 BC), and the Qin-Han Dynasties (221BC - circa third century).[7] To illustrate homoglyphs, we create a reference set for the Shang Dynasty, randomly choosing one character for each concept.

Figure 5 shows representative examples of these homoglyphs, consisting of a character and its five nearest neighbors. The modern character descendant as well as a short description of the ancient concept (which may differ significantly in meaning from the modern descendent) are provided. The description draws upon Li (2012).

Fascinatingly, the homoglyph sets are able to capture related abstract concepts noted in the ar-

[5]https://github.com/dell-research-harvard/HomoglyphsCJK

[6]https://github.com/dell-research-harvard/HomoglyphsCJKTraining

[7]We exclude images from the Shuowen Jiezi - a book on ancient characters - limiting to the most reliable character renders, which were drawn from archaeological sites.

Figure 5: **Ancient Homoglyphs.** This figure shows homoglyph sets constructed for ancient Chinese, with the descendant modern Chinese character and a description of the character's ancient meaning.

chaeological literature. The first line shows that the concepts of writing, law, learning, and morning ("recording the sun") are homoglyphs, and the second line shows that characters for different types of officials are homoglyphs, as are characters denoting "joining." The final line shows that history and government official are homoglyphs - underscoring the central role of the government in writing history - as are characters denoting conquest, tying up, and city center (denoted by a prisoner to be executed by the government, which occurred in the city center).

Not all concepts within each set are related, but many of the connections above have been noted in an archaeological literature examining how ancient peoples conceptualized the world (*e.g.* Wang (2003)). That these meanings can be captured using *vision* transformers is a fascinating illustration of the relationship between images, written language, and meaning in ancient societies.

Another test of extensibility is to compute homoglyphs for all (ancient and modern) characters in Unicode, clustering the embeddings to create homoglyphic sets (termed "Confusables" (Consortium, 2023) by Unicode). Training a homoglyphic space for all of Unicode generally gives sensible results and also has the advantage of being able to measure similarity within a set, something standard Unicode confusables cannot do. An example set, spanning different scripts, is displayed in Figure 6.

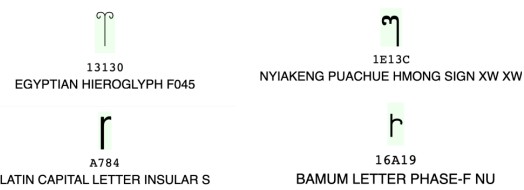

Figure 6: A confusable set across writing systems detected by our model that was trained on all of Unicode

# 7 Limitations

Using homoglyphs for string matching inherits the well-known limitations of string matching. In some cases, OCR destroys too much information for record linkage to be feasible with the resulting strings. Even with clean OCR, sometimes language understanding is necessary to determine the correct match, as in the case of firm names that can be written differently. Homoglyphs do not address other types of string substitutions, like those that result from enumerator misspellings, although in principle a similar contrastive approach could also be developed to quantify other types of string substitutions.

More sophisticated methods have been developed as alternatives to string matching. For example, Ventura et al. (2015) use a random forest classifier trained on labeled data to disambiguate authors of U.S. patents, applying clustering to the resulting dissimilarity scores to enforce transitivity. Arora et al. (2023) develop multimodal record linkage methods that combine the image crops of entities and their OCR. They also develop a vision-only linkage method, which avoids the OCR information bottleneck. Bayesian methods have also been used, *e.g.* Sadinle (2014, 2017). They offer the advantage of uncertainty quantification - another well-known limitation of string matching - but do not scale well.

While these methods offer various advantages, they are not always applicable. Researchers may lack access to the original document images, or may lack the compute or technical resources to process images, limiting the use of OCR-free or multimodal approaches. Language models are less likely to be useful in linking individual or place names, common applications.

Moreover, general purpose, off-the-shelf, end-to-end deep neural methods for record linkage do not exist, *e.g.* the model in Arora et al. (2023) is tuned to a specific Japanese use case. To the extent that labels are required to develop specialized models, the data can be costly to create at a sufficient scale for training end-to-end models. Finally, most social science researchers lack familiarity with deep learning methods but are comfortable processing strings.

For these reasons, off-the-shelf string matching algorithms are often preferred by practitioners and can be the most suitable tool given the constraints. Homoglyphic edit distance integrates information about character similarity from purely self-supervised vision transformers. It can be implemented using our publicly available, simple, off-the-shelf string matching package and is highly extensible, as illustrated by our exercise with characters from ancient Chinese societies. We hope that highly extensible string-matching methods will make the contexts that are feasible for quantitative social scientists to study more representative of the diversity of human societies.

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

## Supplementary Materials

## S-1 HOMOGLYPH Model Details

### S-1.1 Encoder

For both of our applications, we use a DINO pretrained (Caron et al., 2021) vision transformer (ViT) as the encoder. Our implementation of the ViT comes from the Pytorch Image Models library (timm) (Wightman, 2019). Specifically, we use the *vit_base_patch16_224.dino* model that corresponds to the official DINO-pretained ViT-base model with a patch size of 16 and with input resolution of $224^2$. The pretrained checkpoint does not have a classification head.

### S-1.2 Loss function

We use Supervised Contrastive loss (Khosla et al., 2020) as our training objective, as implemented in the PyTorch Metric Learning library (Musgrave et al., 2020), where the temperature parameter is set to 0.1.

### S-1.3 Data Augmentation

We deploy several image augmentations, using transformations provided in the Torchvision library (TorchVision, 2016). These include Affine transformation (only slight translation and scaling allowed), Random Color Jitter, Random Autocontrast, Random Gaussian Blurring, and Random Grayscale. Additionally, we pad the character to make the image square while preserving the aspect ratio of the character render. We do not use common augmentations like Random Cropping or Center Cropping, to avoid destroying too much information.

For augmenting the skeleton of the rendered character itself, we use a variety of digital fonts to render the images. We use 27 fonts for Simplified Chinese, 17 fonts for Traditional Chinese (for both string matching and ancient Chinese), 62 fonts for Korean, and 14 fonts for Japanese.

## S-2 Application-specific details

### S-2.1 Record Linkage

#### S-2.1.1 Data

For each language, the dataset consists of images of characters from the corresponding language rendered with different fonts and augmented during training. The number of characters for each language seen during training is given in Table S-1. Each character can be considered a "class" to which

| Script | Training | Inference |
|---|---|---|
| Japanese | 17,963 | 17,963 |
| Simplified Chinese | 6,621 | 7,806 |
| Traditional Chinese | 8,415 | 8,628 |
| Korean | 3,686 | 3,729 |
| **Total** | **36,685** | **38,126** |

Table S-1: **Training and Inference Sizes**. This table shows the training and inference sizes for different languages.

its digital renders belong. Characters do not need to be seen during training to be considered at inference time, an advantage if users wish to expand the homoglpyph sets (*e.g.* because an OCR engine uses a different character set). We illustrate this empirically by expanding the character set to characters covered by the three OCR engines we explore that were not included in our character ranges used initially for training.

#### S-2.1.2 Batching

**Without hard-negative mining**

Let $\mathcal{B} = 128$ denote the batch size. A batch consists of $m = 4$ views of $\frac{\mathcal{B}}{m}$ characters sampled without replacement. When all the views for a character are utilized, all images are replaced and the sampling process without replacement starts again. "Views" of a character are augmented digital renders using the fonts and transformations described above. One training epoch is defined as seeing the 4 views of all characters exactly once.

**With hard-negative mining**

We find the $k = 8$ nearest neighbors of each character on a checkpoint trained without hard negatives. We do this by rendering all characters in a language with a reference font, Noto Serif CJK font (Tc/Sc/Jp/Ko), chosen for its broad coverage of characters. Each hard negative set includes 4 views of the anchor character, as well as 4 views of each of its $k$ nearest neighbor characters. We randomly intersperse hard negative sets in the batches. One training epoch is now defined as seeing all hard negative sets once. Table S-2 contains the number of epochs we trained each model for.

#### S-2.1.3 Model Validation

80% of characters are used for training, 10% are used to select hyperparameters, and 10% are used

| Model | lr | weight decay | T_0 | T_mult | Epochs |
|---|---|---|---|---|---|
| Japanese - (No HN) distance | 2e-5 | 5e-3 | 1 | 2 | 100 |
| Japanese - (HN) distance | 2e-5 | 5e-3 | 1 | 2 | 30 |
| Simplified Chinese - (No HN) distance | 2e-5 | 5e-3 | 1 | 2 | 30 |
| Simplified Chinese - (HN) distance | 2e-5 | 5e-3 | 1 | 2 | 30 |
| Traditional Chinese - (No HN) distance | 2e-5 | 5e-3 | 1 | 2 | 30 |
| Traditional Chinese - (HN) distance | 2e-5 | 5e-3 | 1 | 2 | 30 |
| Korean - (No HN) distance | 2e-5 | 5e-3 | 1 | 2 | 60 |
| Korean - (HN) distance | 2e-5 | 5e-3 | 1 | 2 | 30 |
| Ancient Chinese - (No HN) distance | 2e-5 | 5e-3 | 300 | 1 | 200 |
| Ancient Chinese - (HN) distance | 2e-5 | 5e-3 | 300 | 3 | 24 |

Table S-2: **Training Hyperparameters**. This table reports the training hyperparameters used for the models. The lr stands for learning rate, weight decay represents the weight decay factor, T_0 is the number of steps until the first restart of the learning rate scheduler, T_mult denotes the factor by which T_0 is multiplied at each restart, and Epochs indicates the total number of training epochs. Parameters not mentioned here use PyTorch defaults. HN denotes offline hard-negative mining.

to select the best checkpoint. We embed the validation images and find the nearest neighbor among the embeddings of digital renders of the universe of characters in the language, rendered with the reference font described above. The top-1 retrieval accuracy is used as the validation metric for the selection of the best checkpoint. We see a peak validation accuracy of 90% for Japanese, 98% for Korean, 91% for Traditional Chinese, and 91% for Simplified Chinese.

### S-2.1.4 Other training details

CJK glyphs are somewhat similar across languages. To converge faster, we initialize the weights of the encoders for Traditional and Simplified Chinese and Korean with the checkpoint used for Japanese, which has the largest number of characters. We use AdamW (Loshchilov and Hutter, 2019) as the optimizer and Cosine Annealing with Warm Restarts (Loshchilov and Hutter, 2016) as the learning rate schedule. We use the standard Pytorch implementation for both. The relevant hyperparameters are listed in Table S-2. We stop training the models once the validation accuracy stagnates and the checkpoint with the best validation accuracy is chosen.

### S-2.2 Homoglyph Sets

We allow for the expansion of the character set beyond what is seen in training because different OCR engines use different character dictionaries (a list of characters supported by the engine). We take the union of characters from the character dictionaries of PaddleOCR, EasyOCR, and EfficientOCR. For each language, we render all its characters using the reference font and embed them using the language-specific HOMOGLYPH encoder. For each character, we then find 800-nearest neighbours (measured by Cosine Similarity between the embeddings) among the set of all renders in the reference set. We store these as a look-up dictionary that contains, for each character in a language, its 800 neighbors and its Cosine Similarity with all of them. This look-up dictionary is used in our modified Levenshtein distance implementation to modify the substitution cost. The dictionaries are available in our GitHub repository.

Table S-1 contains the number of characters that were used to prepare these sets for each language.

### S-2.3 Implementing Homoglyphic Edit Distance

We use a standard algorithm to calculate Levenshtein distance that uses dynamic programming (Wagner and Fischer, 1974). The space and time complexity of the algorithm is $\mathcal{O}(mn)$ where $m$ and $n$ are the lengths of the two strings that are being compared.

We modify this algorithm by switching the standard substitution cost $\lambda$ between two characters $a$ and $b$ with $\lambda * (1 - CosineSimilarity(u(a), u(b))$. Here $u(a)$ and $u(b)$ are the embeddings of the HOMOGLYPH encoder for the language to which $a$ and $b$ belong. $\lambda$ is a tunable hyperparameter but for simplicity, we fix it as 1 for the results shown in the paper. We also fixed the addition and deletion cost as 1 but in the implementation provided in our package and our GitHub repository, the costs are tunable hyperparameters.

### S-2.4 Ancient Chinese Homoglyphs
#### S-2.4.1 Data

The source database (Academia Sinica et al., 2023) from which we collect the ancient Chinese character crops contains 5,024 concepts, comprised of 25,984 character renderings. Each of these concepts is mapped to a modern character. This enables us to insert digital renders of these modern

characters using the same fonts as above (for traditional Chinese) to create more variation. A "class" in this case comprises a character cluster - with both ancient crops and modern digital renders forming the positive samples for a class.

We slightly modify the data augmentation scheme for this application to account for the wide variation in writing styles across centuries. We allow for a slight ($-10$ to $+10$ degree) rotation and also add more transformations tailored to this use case - Random Equalize, Random Posterize, Random Solarize, Random Inversion and Random Erase (randomly erase 0-5% of the image). We apply all augmentations to the digital renders but only apply Random Affine transformation and Random Inversion to the ancient crops.

### S-2.4.2 Batching

We use the same sampling and batching process (this time with a larger batch size $\mathcal{B} = 256$) as we did for the modern homoglyph models. The only difference is in how the hard-negative sets are defined. Instead of one nearest neighbor per concept, for each ancient crop within a concept cluster, we find $k = 8$ nearest neighbors. This gives us as many nearest neighbor sets (hard-negative sets) as ancient crops in our dataset. This allows us to account for the fact that the homoglyphs of a character may differ across different historical periods, spanning millennia.

### S-2.4.3 Model Validation

We use top-1 accuracy as our validation metric, defined as the proportion of correct retrievals of the corresponding modern render (using the reference font Noto Serif CJK Tc) for each ancient image in the validation set. During training, the model reached a peak validation accuracy of 50% demonstrating the difficult nature of this task. We use this metric for selecting the best checkpoint.

### S-2.4.4 Other training details

We again use the AdamW optimizer and Cosine Annealing with Warm Restarts as the learning rate schedule. Relevant Hyperparameters are listed in Table S-2. We stop training when validation accuracy stagnates.

### S-2.4.5 Creation of Ancient Chinese Homoglyphs

The creation of homolgyph sets is analogous to the case of modern characters. Instead of using digital renders from a particular font as the "reference set", we look at the five nearest neighbors of ancient characters within a period. We illustrate homoglyphs using The Shang Dynasty period (1600 BC-1045 BC), the most ancient.

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
