# OpenReview forum: "Quantifying Character Similarity with Vision Transformers"
_EMNLP/2023/Conference — EMNLP 2023 Main_

### Official Review · Reviewer_3GU8 · 2023-08-02

**Soundness:** 3

**Excitement:**

2: Mediocre: This paper makes marginal contributions (vs non-contemporaneous work), so I would rather not see it in the conference.

**Paper Topic And Main Contributions:**

This paper presents a method for matching (hand-written) strings. I think the paper claims that the main contribution of the paper is to use contrastive learning to train a model to compute the similarity between two strings with various fonts. The experimental results show that the method outperforms other similarity metrics for multiple datasets.

**Questions For The Authors:**

A) More clear description about the technical contribution may help understand the paper.

**Reasons To Accept:**

I think the topic is interesting.

**Reasons To Reject:**

The main technical component of the paper is an application of contrastive learning with multiple fonts to learn the similarity between two characters. The way the method uses contrastive learning looks straightforward, so the technical contribution is limited.

Details of string matching over the proposed similarity metric are not provided. This may significantly reduce the reproducibility of the method if it does not publish the code.

Some minor points:

The paper is redundant, repeating similar sentences. This makes the paper hard to read.

The paper claims that the method to train the similarity metric is self-supervised, but for me, this is not self-supervised because it has labels to represent the same characters.

**Reproducibility:**

2: Would be hard pressed to reproduce the results. The contribution depends on data that are simply not available outside the author's institution or consortium; not enough details are provided.

**Reviewer Confidence:**

3: Pretty sure, but there's a chance I missed something. Although I have a good feel for this area in general, I did not carefully check the paper's details, e.g., the math, experimental design, or novelty.

---

> ### Author Rebuttal · Authors · 2023-08-25
>
> We thank the reviewer for the helpful comments and suggestions, and detail our response below.
>
> *”The way the method uses contrastive learning looks straightforward, so the technical contribution is limited.”*
>
> We view the technical contributions of the paper as:
> 1. **Application of SOTA methods to a novel problem**: To the best of our knowledge, we are the first to use Dino-pretrained ViT’s excellent features (Caron et al 2021) to contrastively learn a metric space for character visual similarity. We do so using a state-of-the-art training objective (Khosla et al 2021) and a highly scaleable FAISS inference architecture. While we have not innovated a new backbone or loss function, we have shown how to apply existing methods to a novel setting in a technically rigorous way.
> 2. **Novel data augmentation strategy:** Creating paired data is very costly, and one of the main innovations of the paper is to show that digital fonts can replace costly paired data for training a vision model to encode character visual similarity.
> 3. **Novel edit distance metric**: We modified edit distance - the most widely used metric for record linkage - to incorporate character visual similarity, as measured by a learned metric space. The method can be used off-the-shelf by those who lack familiarity with deep learning frameworks (or coding more generally) in one line of code. It can be cheaply extended to any language, unlike the predominant practice of modifying the edit distance substitution cost with hand-crafted features.
>
> We agree that the contribution is not to improve deep learning in a general sense, but rather to apply state-of-the-art deep learning in such a way that the output can be broadly used, democratizing the benefits of vision transformers for the majority of social scientists who have no familiarity with deep learning frameworks. As mentioned in the paper, end-to-end deep learning approaches fail to generate widespread usage - despite being technically innovative - due to the requirement of in-domain training of models, which requires proficiency with deep learning frameworks. In contrast, with our package users can link two data frames with one line of code, with an API that will be familiar to those users who have experience with statistical packages like R and Stata (which predominate in social science). We feel that the nature of the technical contribution is appropriate for the “Computational Social Science and Cultural Analytics” track.
>
> *”Details of string matching over the proposed similarity metric are not provided. This may significantly reduce the reproducibility of the method if it does not publish the code.”*
>
> We will distribute a package on PyPi for easy application of homoglyphic distance - with no model training required. Unfortunately, we couldn’t share it as a PyPi package to preserve anonymity, but we have shared the code as part of the supplementary materials. We will also provide a public Github Repo for our training and inference code, useful for replicating our indices and extending our results to other scripts and languages. Besides training and inference code, the repo contains implementations of all string-matching approaches mentioned in the paper that will make it simple to re-generate Table 2 for replication. We will also make all of our data publicly available using Hugging Face Hub once the anonymity period is lifted. Unfortunately, we could not share the image data in the submission packet due to file size limitations. These efforts ensure that we fully adhere to EMNLP’s reproducibility checklist.
>
> *”The paper is redundant, repeating similar sentences. This makes the paper hard to read.”*
>
> Based on your and reviewer 3GU8’s comments, we have edited the paper for concision.
>
> *”The paper claims that the method to train the similarity metric is self-supervised, but for me, this is not self-supervised because it has labels to represent the same characters.”*
>
> We understand that the reviewer defines label-supervision as something that disqualifies an approach from the paradigm of self-supervised training, but we would also like to highlight that there are precedents for our usage of self-supervised. Conceptually, rendering characters with different fonts is a form of data augmentation, with augmentation on the skeleton of the characters instead of on the entire image. Contrastive training with a standard augmentation scheme that involves operations like affine transformations, color inversion etc. is widely called self-supervised learning (eg, the seminal contribution SimCLR by Chen at al (2020)), and our font-based augmentations are analogous. We simply use instructions for how to render fonts, rather than other image augmentation libraries, to create these augmentations. Just like in standard image augmentations, we generated the labels for free, with no need for human annotations. Semantics aside, the main fact of practical importance is that the method can be extended to virtually *any* Unicode character without the need to create labels, making it highly extensible. We agree that defining self-supervised learning is a tricky task in the context of contrastive learning and thank the reviewer for providing us the space to discuss this.

---

### Official Review · Reviewer_BJgQ · 2023-08-04

**Soundness:** 4

**Excitement:**

4: Strong: This paper deepens the understanding of some phenomenon or lowers the barriers to an existing research direction.

**Missing References:**

Give some reference for Soundex.   (Yes, I realize that it might be hard to find a primary reference.   I couldn't do that quickly.   Perhaps a secondary source?)



**Paper Topic And Main Contributions:**

This paper presents a novel algorithm for matching strings in OCR text for languages using CJK script.   The intended application is linking instances of the same entity between different historical documents.   So OCR is necessary and character recognition errors are a significant problem.

The algorithm uses renderings of each character in several digital fonts as positive examples for contrastive training of an algorithm to embed renderings of characters in a vector space.   (This is somewhat similar to EfficientOCR.)   Cosine distance in this space (between two renderings in a fixed standard font) is used as the substitution penalty in the string matcher's Levenshtein distance computation.   In other words, the substitution penalty is less if the characters look similar.   This is a simple combination of existing methods but it seems novel and well placed.

The algorithm is tested on two small hand-curated datasets of Japanese firm names:   1092 pairs of customer-supplier entries in one case (two different OCR engines) and then 238 firm names to be located in a very large list of firm names (a third OCR engine).  Simulated ground truth data was created for four large datasets of place names by running each through two different OCR engines and seeing how well the matcher could correctly pair the output images.   This is a nice job of combining simulated and manually curated data in a situation where it could be expensive to get large amounts of gold standard data.

They also run their embedding algorithm on a dataset of images of ancient Chinese characters.  There is no formal evaluation of performance, but the resulting nearest neighbor sets give a sense of the algorithm's ability to find similar-looking characters in a less well controlled dataset.   Also this is fun.


**Questions For The Authors:**

How many digital fonts were used in training the embedding model?

Were the ancient Chinese characters written on something like paper or ceramics, or were they carved?   (Just curious.)

P. 5 column 2 right near the bottom:   What was the actual accuracy of the end-to-end neural network methods?

**Reasons To Accept:**

The method is simple but looks very useful and straightforward to reproduce.   A quick reference (p. 3 column 1 second paragraph) suggests they have a package for distribution.

The new method outperforms previous string matching algorithms on this task.   They make the believable claim that previous methods (e.g. end-to-end neural networks) may have somewhat higher accuracy but are costly to (re-)tune for each use case.

The paper is mostly clear and well written, with some exceptions noted below.

**Reasons To Reject:**

The idea isn't rocket science (but I do think it's nice).

The paper is somewhat repetitive.   The content seems a bit short for a long paper.   Then again, I think it would be quite difficult to squeeze it into the length of a short paper.   I guess it just needs to be a short long paper.

The abstract and the limitations/conclusions section have appropriate content but need editing (see below).

**Reproducibility:**

4: Could mostly reproduce the results, but there may be some variation because of sample variance or minor variations in their interpretation of the protocol or method.

**Reviewer Confidence:**

3: Pretty sure, but there's a chance I missed something. Although I have a good feel for this area in general, I did not carefully check the paper's details, e.g., the math, experimental design, or novelty.

**Typos Grammar Style And Presentation Improvements:**

Make a table of the key facts for all datasets, e.g. language, how many strings to be matched, which OCR engines used.   And give each dataset a short name.   That will allow you to refer to them easily and not end up with so much duplicative description of the datasets.

Try to reduce the repetition between different sections of the paper, e.g. the introduction and section 2.2 and section 6.

Notice that the limitations section does not count towards the 8-page limit.   So you have space to add examples (e.g. some of the linked text from the first two experiments) and/or make the algorithm easier to follow.

Perhaps an architecture diagram for the main algorithm in section 2.1?    When I got to the third paragraph in the second column of page 3 ("To compute character's similarity") I had to re-read a couple times to see how this part fit together with what you had just done in the first two paragraphs.   2.1 is the critical main part of your algorithm, so use the available page space to make it very very clear and easy to follow.

In the early parts of section 3, remind us of the language for the first two datasets.

The abstract needs to be rewritten.   Shorten it and divide into 2-3 paragraphs.   Some key definitions (e.g. ViT) should be repeated in the main text.

The paper doesn't have a conclusions section.   The limitations section repeats too much from earlier sections and some of its material feels like it belongs in the missing conclusions.   Also, mention the fact that the test datasets are small.   Everyone will understand why but it is definitely a limitation and perhaps some future author will be inspired to make more gold standard data for this task.

"Image render" ---> "image rendering"

Table 2:  boldface the best result number

---

> ### Author Rebuttal · Authors · 2023-08-25
>
> We thank the reviewer for their detailed and thoughtful comments. We respond to the reviewer's questions and concerns below:
>
> *“The idea isn't rocket science (but I do think it's nice).”*
>
> Response: We agree that the main contribution is to take existing tools (DINO ViT embeddings, supervised contrastive loss) and apply them to solve a novel problem, versus innovating on the architectures themselves. We wanted this technology to be accessible to those with limited coding experience, who form the bulk of potential users, and a simple architecture greatly facilitates this. Our homoglyphic edit distance package links data frames with a single line of accessible code.
>
> *”The paper is somewhat repetitive.”*
>
> Respnose: We thank the reviewer for the helpful suggestions on how to make the writing clearer and more effective. We have followed the reviewer’s advice, editing for concision and clarity.
>
> *”How many digital fonts were used in training the embedding model?”*
>
> Response: We use 62 open-source fonts for Korean, 27 for Simplified Chinese, 17 for Traditional Chinese,  and 13 for Japanese. Our training budget required us to base the number of fonts for each script on the total number of characters in that script, using more fonts for characters with a smaller Unicode range.
>
> *”Were the ancient Chinese characters written on something like paper or ceramics, or were they carved?”*
>
> Response: According to Tsuen-hsuin Tsien’s seminal work on ancient Chinese writing, *Written on Bamboo and Silk: The Beginnings of Chinese Books and Inscriptions* (2004), characters written during the Shang period tended to be discovered archaeologically from inscriptions on bone tortoise shells, ox bones, or bronze wares. Several archaeological findings suggest that such inscriptions would first be written with brush pens with inks and later carved with knives or other tools.
>
> *”P. 5 column 2 right near the bottom: What was the actual accuracy of the end-to-end neural network methods?”*
>
> Response: As we now clarify, Arora et al. (2023) achieve an accuracy of 87.8 with Vision-only contrastive training with a Vision Transformer backbone and a peak accuracy of 94.5 using a Language-Image model (CLIP) backbone with multimodal pooling, trained specifically on the train split of this dataset and hence not extensible zero shot to other applications.
>
> *Soundex reference*
>
> Response: While we have likewise never seen an original paper describing Soundex, we now cite the patent: Russell, Robert C. U.S. Patent #US1261167A and point readers seeking a readily accessible description to U.S. Census Bureau (2007), “Soundex System: The Soundex Indexing System.” https://www.archives.gov/research/census/soundex
>
> *”Make a table of the key facts for all datasets, e.g. language, how many strings to be matched, which OCR engines used. And give each dataset a short name.”*
>
> Response: We have made a table with key facts for all the datasets and also given them a short name. Here is a markdown-friendly version:
> | Task                            | Query Sample Size | Target Sample Size | Dataset Name   |
> |---------------------------------|-------------------|--------------------|----------------|
> | Synthetic Japanese              | 86,470            | 86,470             | synth-ja       |
> | Synthetic Korean                | 48,809            | 48,809             | synth-ko       |
> | Synthetic Traditional Chinese   | 66,943            | 66,943             | synth-zht      |
> | Synthetic Simplified Chinese    | 20,162            | 20,162             | synth-zhs      |
> | Historic Japanese Companies     | 238               | 68,352             | historic-japan |
>
> *”Perhaps an architecture diagram for the main algorithm in section 2.1?”*
>
> Response: We agree that this will be helpful. EMNLP does not provide us with a mechanism to share this now but we will add it to the paper.
>
> *Various suggestions to streamline and clarify the writing*
>
> Response: We greatly appreciate these helpful suggestions and have incorporated them into the paper.

---

### Official Review · Reviewer_Qkwc · 2023-08-05

**Soundness:** 4

**Excitement:**

4: Strong: This paper deepens the understanding of some phenomenon or lowers the barriers to an existing research direction.

**Paper Topic And Main Contributions:**

The paper proposes a novel approach to record linking based on string similarity as measured by edit distance which builds on existing methods which contemplate that not all edits or substitutions are equally probable.  The focus is on visual similarity between characters, and to this end the authors develop a self-supervised embedding of images of characters based on vision transformers, such that edit distance can be weighted by visual similarity.

**Reasons To Accept:**

The approach is novel, well-argued, and effective, and addresses several inherent difficulties of the problem
- Training data are reasonably augmented to focus on the composition & structure of characters rather than style of writing / font
- The authors overcome a paucity of evaluation data by synthesizing their own, but they do provide some evaluation on some real data that are available — this may be a limitation of the evaluation, however the methods for data synthesis proposed seem sound and it seems likely results on a synthetic evaluation will generalize to real data
- Limitations are clearly discussed and accurate, and the authors acknowledge that there are more sophisticated end-to-end deep neural methods that may solve the record linking problem more directly and decisively, and incorporate language understanding into the process, but these methods may not always be practical.

**Reasons To Reject:**

This is a well-written paper which clearly and effectively argues for a novel and interesting approach, providing an adequate evaluation of the approach and a clear discussion of its limitations — this reviewer sees no clear or overwhelming reason to reject.

One limitation not discussed in the paper (edit: the authors have, in their rebuttal, agreed to revise the paper to explicitly address this) is that these experiments focus on a single script, and it’s not demonstrated how the approach would generalize to other alphabets, but the authors picked a good example of a high cardinality alphabet where the limitations of an unmodified edit distance in record linking would be most pronnounced.

**Reproducibility:**

4: Could mostly reproduce the results, but there may be some variation because of sample variance or minor variations in their interpretation of the protocol or method.

**Reviewer Confidence:**

4: Quite sure. I tried to check the important points carefully. It's unlikely, though conceivable, that I missed something that should affect my ratings.

---

> ### Author Rebuttal · Authors · 2023-08-25
>
> We thank the reviewer for their insightful comments. Below, we provide responses to their questions and concerns.
>
> *One limitation not discussed in the paper is that these experiments focus on a single script, and it’s not demonstrated how the approach would generalize to other alphabets, but the authors picked a good example of a high cardinality alphabet where the limitations of an unmodified edit distance in record linking would be most pronnounced.*
>
> Response: The reviewer highlights exactly our reasoning for focusing on the CJK script, and we now explicitly acknowledge this limitation. We provide well-documented and easy-to-implement code to facilitate the extension of our approach to other scripts. We do not provide evaluations for other high cardinality scripts primarily because our research team does not have the expertise or budget to create gold quality record linkage data for these scripts, and to our knowledge no benchmarks currently exist. We do hope this work will encourage other researchers to develop such evaluations.
>
> We have tried an exercise computing homoglyphs for all of Unicode; e.g., embedding all of Unicode. This yielded generally sensible results - e.g., Unicode characters 78128, 42884,123196,92697 (unfortunately, these Egyptian Hieroglyph, Latin, Nyiakeng, and Bamum characters will not display properly in OpenReview, and EMNLP prohibits us from posting links to an image). We didn’t pursue this further because there was no ground truth data to use for evaluation. However, - along with the Ancient Chinese exercise in the paper - it suggests that the approach produces sensible results across a wide variety of (ancient and modern) scripts. We can include some of these universal unicode confusable sets in the supplemental materials.

---

### Meta-Review · Area_Chair_eKMw · 2023-09-19

**Recommendation:** 4

**Metareview:**

The paper proposes a new technique for record linkage that uses the visual features of characters, rather than simple edit distance, arguing that this addresses a common problem in OCRed historical text. It uses a clever and novel combination of existing tools (DINO ViT, contrastive loss) to solve a real problem that applied researchers have. While the paper does not make a novel contribution in loss function or model architecture, the combination of tools and the task framing are novel and the paper will be useful.

One reviewer raised concerns about replicability or whether the process will be easy to use, but another reviewer and the authors point out that the pipeline will be available as a Python package, which will be useful to applied researchers.

The reviewers find that the paper is somewhat repetitive at times and an architecture figure would be helpful, but the authors have agreed to revise the paper to address these issues.

Reviewers raise a number of clarification questions, comments about terminology, and requests for minor extensions/additional citations. The authors thoughtfully respond to each of these and propose changes to the paper that will address them.

---

### Decision · Program_Chairs · 2023-10-07

**Decision:**

Accept-Main

**Comment:**

The paper proposes a new technique for record linkage that uses the visual features of characters, rather than simple edit distance, arguing that this addresses a common problem in OCRed historical text. It uses a clever and novel combination of existing tools (DINO ViT, contrastive loss) to solve a real problem that applied researchers have. While the paper does not make a novel contribution in loss function or model architecture, the combination of tools and the task framing are novel and the paper will be useful.

One reviewer raised concerns about replicability or whether the process will be easy to use, but another reviewer and the authors point out that the pipeline will be available as a Python package, which will be useful to applied researchers.

The reviewers find that the paper is somewhat repetitive at times and an architecture figure would be helpful, but the authors have agreed to revise the paper to address these issues.

Reviewers raise a number of clarification questions, comments about terminology, and requests for minor extensions/additional citations. The authors thoughtfully respond to each of these and propose changes to the paper that will address them.